# Exercise Levels and Preferences in Cancer Patients: A Cross-Sectional Study

**DOI:** 10.3390/ijerph17155351

**Published:** 2020-07-24

**Authors:** Alice Avancini, Valeria Pala, Ilaria Trestini, Daniela Tregnago, Luigi Mariani, Sabina Sieri, Vittorio Krogh, Marco Boresta, Michele Milella, Sara Pilotto, Massimo Lanza

**Affiliations:** 1Department of Medicine, Biomedical, Clinical and Experimental Sciences, University of Verona, 37134 Verona, Italy; alice.avancini@univr.it; 2Department of Research, Epidemiology and Prevention Unit, Fondazione IRCCS Istituto Nazionale dei Tumori, 20133 Milan, Italy; sabina.sieri@istitutotumori.mi.it (S.S.); vittorio.krogh@istitutotumori.mi.it (V.K.); 3Medical Oncology Unit, University of Verona, Azienda Ospedaliera Universitaria Integrata, 37134 Verona, Italy; ilariatrestini92@gmail.com (I.T.); danielatregnago@libero.it (D.T.); michele.milella@univr.it (M.M.); sara.pilotto@univr.it (S.P.); 4Department of Medical Statistics, Biometry and Bioinformatics, Unit of Clinical Epidemiology and Trial Organization, Fondazione IRCCS Istituto Nazionale dei Tumori di Milano, 20133 Milan, Italy; luigi.mariani@istitutotumori.mi.it; 5Department of Computer, Control and Management Engineering Antonio Ruberti, Sapienza University of Rome, 00185 Rome, Italy; marco.boresta@uniroma1.it; 6Department of Neurosciences, Biomedicine and Movement Sciences, University of Verona, 37134 Verona, Italy; massimo.lanza@univr.it

**Keywords:** exercise, cancer, preferences, health promotion, adherence

## Abstract

Background: Despite the benefits related to physical exercise, large numbers of cancer patients are not sufficiently active. Methods: To investigate exercise levels and preferences in cancer patients, a cross-sectional study was conducted on a random sample of 392 cancer outpatients who anonymously completed a questionnaire investigating general and medical characteristics, and expressed willingness to participate in exercise programs. Current exercise levels were estimated with the Leisure Score Index (LSI). Results: Most patients (93%) were insufficiently active but 80% declared an interest in exercise programs. Patients preferred oncologist-instructed programs and specified particular exercise needs. Multivariate logistic regression showed that willingness to exercise was associated with education (OR: 1.87; 95% CI: 1.15–3.04 beyond age 14 years vs. up to 14 years) and current physical activity (OR: 1.92; 95% CI: 1.92–3.63 for sweat-inducing activity >2 times/week vs. <1 time/week). Patients given chemotherapy were less inclined to exercise (OR: 0.45; 95% CI: 0.23–0.86) than those who did not. LSI was lower if cancer stage was advanced (β: -0.36; 95% CI: −0.75 to −0.02) than if it was in remission. High LSI was also associated with longer education, lower BMI, and longer time after diagnosis. Conclusion: Cancer patients are insufficiently active but are willing to participate in personalized exercise programs. Information from this survey may help in designing personalized interventions so these patients will achieve sufficient exercise.

## 1. Introduction

In 2019 it was estimated that about 3.5 million Italians (5.3% of the entire population) are living after a cancer diagnosis [1]. Improvements in medical treatments have led to a substantial increase in the proportion of cancer patients with death rates similar to those of the general Italian population [2].

Cancer and its treatments are associated with various side effects that negatively affect the patient’s quality of life for a long time after the conclusion of therapies [3,4]. There is growing evidence that in cancer patients (especially breast, colon and prostate) [5,6,7] an active lifestyle is associated with a lower risk of recurrence and mortality. Physical activity (PA) refers to any bodily movement produced by skeletal muscles that requires energy expenditure [8]. Exercise is defined as a subcategory of PA, consisting of structured, planned and repetitive movement [8]. Exercise was shown to be safe and feasible in oncological settings [9] and several studies found that exercise improved patients’ quality of life during [10,11] and after treatment [11]. Positive effects of exercise include increasing cardiorespiratory fitness [12] and muscular strength [13], and improvement in body composition [13]. Additionally, exercise helped regulate several side effects of cancer treatment, such as fatigue [14], and nausea [15], and improved the psychological status, for instance, reducing levels of anxiety and depression [16]. 

Despite the benefits related to PA and exercise, a large percentage of cancer patients from 25% to 84% are not sufficiently active [17,18,19] and the level of exercise has been seen to decrease after cancer diagnosis [20]. A multitude of factors influence the participation of the general population in exercise programs (e.g., lack of time, cost, logistic difficulties, etc.) [21]. Cancer patients face further obstacles on account of their condition (e.g., cancer-related fatigue, muscle weakness, nausea, sleep disorders) [22,23]. To develop a successful exercise intervention, cancer patients’ barriers and preferences must be considered, allowing them to pick the activities they perceive as beneficial and enjoyable [23,24,25,26]. International studies investigated the preferences and determinants of exercise levels in cancer patients and survivors [18,27,28,29,30,31,32,33], but data on the Italian population are lacking. Furthermore, cultural differences in this area might be significant. In order to overcome this information gap the STIP-ON (Sustainable training in pazienti oncologici) survey was designed with the following aims: (i) To understand the size of the problem, i.e., to calculate the prevalence of insufficient exercise among cancer patients; (ii) to analyze the patients’ characteristics associated with insufficient exercise; (iii) to analyze the patients’ characteristics associated with their motivation/willingness to take part in a future intervention program on exercise; (iv) to describe patients’ preferences about exercise.

The rationale of the study is that understanding patients’ preferences and barriers to physical activity will make it easier for them to participate successfully in a future intervention study to improve their physical fitness.

## 2. Materials and Methods

### 2.1. Study Design and Participants

This STIP-ON study is a cross-sectional survey. Data were collected and recorded anonymously from patients visiting the cancer outpatients’ facilities at the Oncology Unit of “Azienda Ospedaliera Universitaria Integrata”, University of Verona, Verona, Italy between July 2018 and April 2019. Cancer patients’ eligibility criteria were: age ≥18 years, a cancer diagnosis and adequate Italian language proficiency to answer the survey questionnaire (QEX). Invited participants included all kinds of cancer survivors (including those whose diagnosis had just been made or was being defined). The STIP-ON sample was thought to be representative of patients visiting the cancer outpatients’ facilities: on randomly selected days they were approached face to face, informed about the study and asked whether they would be willing to complete the questionnaire anonymously to investigate their characteristics and preferences regarding exercise. Invited participants were systematically asked by the staff if they had already completed the survey another time/day before this QEX was administered. A duplicate check was done, looking for duplicates by date of birth, province of residence, sex, education, and marital status. If interested in participating, patients were asked to give signed informed consent, and received a leaflet (Appendix A) describing the purpose of the study and a copy of QEX. QEX was completed on the spot or could be taken home and returned within a week. In both cases, participants were asked to leave the completed QEX anonymously in a special ‘ballot box’. 

The project was reviewed and approved by the Ethics Committee for Clinical Trials (Prot. No. 49018), University of Verona. All study procedures were conducted following the last revision of the declaration of Helsinki as well as the declaration of Oviedo. The study protocol was designed to adhere to Good Clinical Practice principles and procedures and had to comply with Italian legislation.

### 2.2. The Survey Questionnaire (QEX)

The QEX is a self-administered survey to collect cancer patients’ preferences and characteristics associated with exercise. The questionnaire is the result of a co-design process that involved patients (via patients’ associations) and experts, including oncologists, kinesiologists, epidemiologists and psycho-oncologists. The pilot version of QEX was developed based on a literature review [27,28,29,30,31] made available to these ‘reviewers’ to give feedbacks and make an unofficial peer review to develop the current version. While QEX is a self-reported, anonymous survey, staff support (including dedicated personnel in the room) was available during the survey to address any questions. The QEX comprises 31 items (Q1–Q31), divided into four sections: (a) General characteristics (from Q1 to Q9); (b) Physical exercise level (from Q10 to Q11); (c) Physical exercise preferences (from Q12 to Q26); (d) Cancer diagnosis and treatment (from Q27 to Q31). A copy of QEX is available online as Appendix A.

#### 2.2.1. Questions 1–9: General Characteristics

The following demographic, anthropometric and socio-economic characteristics of patients are collected in the QEX: birth date (day, month, year), sex, province of residence, education level (elementary—up to age 10–11 years/secondary—up to 14 years/secondary—up to 18–19 years/college–university/postgraduate), marital status (single/married/divorced/widowed, occupational status (retired/homemaker/part-time employed/full-time employed/other), perceived economic adequacy (inadequate/barely adequate/adequate/more than adequate), body weight (kg) and height (cm) (both continuous). Age was calculated by subtracting the date of birth from the date of QEX compilation and classified in two categories (<65; ≥65y). Body mass index (BMI) was calculated from the weight in kilograms divided by the height in meters squared (kg/m^2^). BMI categories were defined as follows: underweight (BMI <18.5 kg/m^2^), normal weight (BMI 18.5–24.9 kg/m^2^); overweight (BMI 25–29.9 kg/m^2^); obese (>29.9 kg/m^2^) [34].

#### 2.2.2. Questions 10–11: Level of Physical Exercise

The QEX inquiry about current exercise level was based on questions from the Godin Leisure-Time Exercise Questionnaire (GLTEQ) [35] which is widely used for cancer patients [36]. A detailed description of the computation of LSI from GLTEQ is found elsewhere [35,36]. In brief: (i) The GLTEQ enquires about the previous week’s leisure time frequency (times/week) of vigorous, moderate- and mild-intensity exercise; (ii) Each exercise intensity is associated with the metabolic equivalent of the task (MET): MET = 9 for vigorous, MET = 5 for moderate, MET = 3 for mild intensity exercise [35]; (iii) The LSI is then calculated as the sum of (vigorous × 9) + (moderate × 5) per-week exercise frequency according to Godin and Shepard [35]. Based on their LSI, patients are classified as active (if LSI ≥24) or insufficiently active (if LSI <24) according to the 2010 release of American College of Sports Medicine (ACSM) Exercise Guidelines for cancer patients [37]. The ACSM guidelines suggest cancer patients engage in at least 150 min/week of moderate or 75 min/week of vigorous exercise [37]. The QEX includes an additional self-rated question about the frequency (times/week) of sweat-inducing activity. There are three categories of frequency (often/sometime/never–rarely) these questions and categorization are also taken from GLTEQ [35].

#### 2.2.3. Questions 12–26: Physical Exercise Preferences

Exercise preferences were investigated by questions from previous studies [27,28,29,30,31]. The first question concerns the patient’s willingness to participate in an exercise program (yes/no/maybe). Respondents were asked about their preference regarding: who would give them exercise instructions (oncologist/nurse/kinesiologist/nutritionist/physiotherapist/another cancer patient/no preference/other); how to receive exercise instructions (face to face/by telephone/videotape/television/brochure-pamphlet/over the internet/no preference/other); with whom they would prefer to exercise (nobody/other cancer patients/family members/friends/a group/no preference/other); where (at home/at a community fitness center/at an adapted exercise fitness center/outside/no preference/other); what time of day (morning/afternoon/evening/no preference); what part of the week (weekday/weekend/no preference) and how often (from never to seven times/week). Further information was collected on preferred intensity (mild/moderate/strenuous/no preference), session content variability (same each time/different each time/no preference), “helper” during the program (nobody/exercise specialist/neighbor/colleague/friend/son-daughter/spouse/other relative), supervision (unsupervised/supervised/no preference) and kind of exercise program (individual with a program to follow at home/individual with personal trainer/in a group with a kinesiologist/physiotherapist/exercise specialist). There were also two open-ended questions in which respondents were encouraged to list the top three preferred exercise activities in winter and summer.

#### 2.2.4. Questions 27–31: Cancer Diagnosis and Treatment

Medical variables were self-reported by patients and included: tumor site (lung/colorectal/breast/head-neck/upper gastrointestinal/gynecological/urogenital/melanoma/ hematological/other), disease status (unknown/in remission-cured/early/advanced/metastatic), date of diagnosis (month/year), type of treatment (surgery/chemotherapy/radiotherapy/hormone therapy/other) and current treatment status (about to start/ongoing/completed/not known).

Time from diagnosis was calculated by subtracting the date of diagnosis from the date of QEX compilation and was classified in two categories using the median (≤30 months; >30 months).

### 2.3. Statistical Analysis 

Descriptive analyses are presented as mean, medians and IQR for continuous variables and frequencies and percentages for categorical variables. Categorical non-ordinal variables were incorporated as dummy variables (Xd) in regression models so that Xd = 1 if the condition is true and Xd = 0 if not. Minimally adjusted models to investigate patients’ characteristics associated with willingness to participate and current exercise level included age and sex as explanatory variables. Multivariable regression models to investigate patients’ characteristics associated with their willingness to participate and current exercise level included explanatory variables, selected in advance, in the fiducial model that subsequently maximized the goodness of fit, according to the Akaike information criterion (AIC) [38]. These variables included: sex, age, education, residence, perceived income adequacy, marital status, occupational status, frequency of sweat-inducing activity, tumor site, disease, chemotherapy, surgery, radiotherapy, hormone therapy, other treatments, treatment status, time from diagnosis, “lack of preference” (score 0 for no no-preference reply, score 1 for 1 no-preference reply, score 2 for 2 or more no-preference replies to exercise preference questions), “independence” (score 0 if “on my own” never chosen in exercise preference questions, score 1 otherwise). 

The sample size of 200 cancer patients was based on the feasibility criteria of the study. The expected sample allowed estimates of binary variables [e.g., percentages of active (*p*) vs. percentages of insufficiently active (*P* = 1 − *p*) or percentages of patients expressing interest vs. percentages expressing no strong interest] with a standard error of 0.035 and a confidence interval between 0.43 and 0.57, assuming the most unfavorable proportion equal to 0.5 (*P* = 0.5) and alpha 5%.

Statistical tests were two-sided and *p* values < 0.05 were considered significant. The Stata statistical package, version 14 (Stata Corp, Texas, TX, USA) was used.

## 3. Results

The flow diagram of participants is shown in Figure 1. Among the 694 patients approached, 249 (36%) declined to participate in the survey. The most frequent reason for declining was lack of interest. Among the 445 who agreed to participate, 53 did not return the QEX, leaving the final study sample of 392 subjects (55% of the patients approached).

### 3.1. General and Tumor Characteristics

Demographic and medical variables stratified by the willingness to participate in the exercise program are set out in Table 1. The participants’ mean age was 59.6 ± 12.2 y, 61% were female, 69% were married and 61% had at least higher education, up to age 18–19 years. Overall, 83% of participants were still on active treatment; the most frequent tumor sites were upper gastro-intestine (42%) and breast (26%), with a mean time from diagnosis of 2.4 years.

### 3.2. Exercise Behavior 

Details on participants’ exercise behavior by sex and age are shown in Table 2. Patients reported mean frequencies of strenuous, moderate and mild exercise of 0.2 ± 0.84; 0.71 ± 1.43 and 1.56 ± 2.15 times/week respectively. The LSI found 93% of patients insufficiently active, and only 7% met physical activity recommendations [33]. Men and women reported similar exercise behavior through age. Older patients (≥65 years) reported a decline in strenuous and moderate exercise frequencies and an increase in mild exercise compared to <65 years.

### 3.3. Exercise Preferences

Participants exercise preferences are listed in Table 3. Overall, 80% of the respondents were willing (i.e., yes or maybe) to participate in an exercise program designed for cancer patients. Over half (57%) preferred to receive exercise instructions from an oncologist, about 30% from a physiotherapist and 20% from a kinesiologist. The preferred way to receive exercise instructions was with a face-to-face approach (72%), followed by no preferences (12%). The people they preferred to exercise with were other cancer patients (27%). The favorite place for exercise was outside (27%), followed by an adapted exercise fitness center (22%) or at home (21%). Almost half (48%) indicated they preferred exercising in the morning and 70% preferred exercising during a weekday. Just over a third (37%) opted to exercise twice a week and another 30% three times a week. Walking, swimming and biking were the favored activities in summer, while in winter participants opted for walking, gym training and swimming. Participants also specified that they preferred training at mild (48%) or moderate (39%) intensity. About 34% of patients preferred exercise sessions to vary. Most of them (62%) preferred supervised exercise. The preferred helpers were spouses (28%), exercise specialists (22%) or friends (19%). The preferred exercise program was in a group with an expert (40%). 

### 3.4. Relations between Demographic/Medical Variables within Exercise Behavior and Willingness to Participate in Exercise Program

Table 4 shows the relations between characteristics of cancer patients willing to participate the exercise program. Multivariable logistic regression models showed that these patients most likely attended at least secondary school beyond age 14 years (OR = 1.87, 95% CI = 1.15 to 3.04) and had more than double the sweat-inducing activity per week (OR = 1.92, 95% CI = 1.92 to 3.63). Among medical treatments, patients who received chemotherapy were less willing to participate (OR = 0.45, 95% CI = 0.23 to 0.86) than those who did not.

Table 5 shows how patients’ characteristics were related to current exercise levels. Levels were lower in patients with BMI ≥25 (β = −0.33, 95% CI −0.57 to −0.10) than those with BMI <25. Exercise levels were higher in patients who had attended at least secondary school beyond age 14 years (β = 0.32, 95% CI 0.09 to 0.55) compared with those with less than secondary school. Patients who self-defined their disease stage as “advanced” had lower exercise levels (β −0.36, 95% CI −0.75 to −0.02) than those in remission/cured. 

## 4. Discussion

The STIP-ON survey found that only 7% of cancer patients do enough physical exercise. Previous studies reported the percentage of cancer patients with adequate exercise levels, between 16–85% [17]. Considering the impact of physical inactivity on the quantity [5,6,7] and quality [10,11] of life in cancer patients this is an alarming result.

Roughly 80% of patients were willing to start an exercise program designed for cancer patients. Previous studies reported similar results, finding that the majority of bladder [39], non-Hodgkin’s lymphoma [29], prostate [27], head and neck [31], endometrial [28], ovarian [30] and breast [27] cancer survivors were interested in an exercise program. This is important because it supports the cancer patients’ desire for an exercise service.

Several socio-demographic characteristics were associated with the willingness to participate in an exercise program. Willingness decreased with age, also in fully adjusted models, and this was to be expected given the growing difficulties and comorbidities due to aging. Age has been associated with low adherence to exercise in cancer patients in various studies [30,40]. What is interesting is that, even among the older patients, more than two-thirds said they might be interested in taking part in an exercise program. Women were more willing to participate than men. That was found in all models, even after adjustment for medical and socio-demographic variables. That women cancer patients adhere better than men in exercise programs is suggested by an intervention study in rectal cancer patients [40] although a systematic review evaluating the predictors of adherence to exercise interventions during cancer treatment suggested that adherence was best among men [41]. Better-educated patients were more willing to participate. This was reported in other studies, too [30,42], and a likely explanation is well-educated people’s greater awareness and knowledge of the benefits of exercise. It is interesting that economic security was not related to the willingness to participate, and that too was suggested by other studies [40]. This lack of association might be the result of two concomitant and opposing phenomena: those who have less financial availability willingly accept a free offer to exercise; the same poorer people, however, may have less desire to exercise because they are less motivated or because they do manual work. Patients who reported higher frequencies of sweat-inducing activity were more willing to participate in an exercise program that those less frequently reporting it. This can be summed up with the Italian saying: “it rains where it’s already wet”; in other words, those who are most motivated are those who would need it less. No similar results were found in the literature, but a possible explanation is that those who have already done more physical exercise perceive the benefits better and are therefore more ready to improve or increase their level [43]. Chemotherapy was inversely associated with the willingness to participate. There is one study that found no relation between cancer treatment and adherence in high-intensity and low-to-moderate-intensity exercises [42]; other studies found chemotherapy [41] and its side effects [22] were associated with low adherence to physical exercise programs. One explanation for these contradictory results may be that chemotherapy is a generic term that includes different drugs and various possible side effects. There were no differences in willingness to participate based on other medical variables, and this is consistent with previous work on this topic [30].

Regarding the preferred source of exercise instruction, the oncologist was the preferred person to deliver instructions in the present survey and this is not in line with the current standard of care. Previous investigations reported an exercise expert (kinesiologist) as the favorite [17]. Findings from the present survey may be related to the lack of exercise specialists for patients at the Verona Hospital Oncology Unit. The trusting relationship between the patient and the oncologist built up during the cancer journey is another likely explanation. Less than half of oncologists promote exercise with their patients [44]. Barriers that interfere with exercise promotion by oncologists were identified as lack of time, limited access to an exercise specialist/program and lack of knowledge about exercise in cancer [45]. However, educational sessions about exercise in cancer patients and caregivers, specific education materials (leaflets, brochures, posters, etc.) and/or a kinesiologist as part of the clinical team are recognized factors to help promote exercise [45].

Social support plays a role in exercise program compliance [46]. In the oncological setting, social support enhances emotional well-being [47] and is related to PA engagement [48]. The present results are in line with this: 55% of patients preferred exercise with others (cancer patients, relatives, friends); about 87% expressed interest in having a helper, i.e., a person to help and motivate them with the exercise, identifying various subjects: the spouse or other relatives, or exercise specialists. Social support from different helpers has been seen to be effective for behavior change [49]: family, friends, peers, exercise specialists, healthcare providers, and other influential subjects might be the key figures to support compliance and the maintenance of exercise over time [50].

Although in previous studies there was a marked preference for a home-based program [17,27,28,30,39], in this study similar percentages of patients preferred exercising outside, or in an adapted exercise fitness center, or at home. This suggests that providing different program options would boost compliance for exercise interventions. To reinforce this assumption, subjects were asked what they would choose out of three exercise options (individually with a program to follow at home/individually in a gym with a kinesiologist/in a group class with a kinesiologist/none of these). More than 90% indicated their preference among these options.

The majority of STIP-ON participants preferred a supervised exercise program. This finding contrasts with studies on bladder [39], head and neck [31], prostate and breast [27] cancer, but is in line with other investigations on mixed [51], lung [52] and endometrial [28] cancers. One explanation might be related to the patients’ health condition: cancer-related treatments affect normal physical function and influence daily activities, hence the need for supervision from a qualified figure to avoid adverse effects. Moreover, supervised exercise intervention may give additional benefits for cancer patients. A recent metanalysis including a total of 4519 patients with mixed cancer types evaluated the effect of exercise on quality of life and physical function; it found twice the effect size for supervised compared to unsupervised training [53].

In line with previous studies [27,28,30,31,39,51], a substantial proportion of patients indicated walking as their favorite activity, in winter and summer. Walking programs have been effective to manage treatment side effects and improve physical functions in cancer populations [54,55]. Walking is relatively safe, flexible and easy as it does not require special skills [56]. Moreover, walking can be done in different environmental situations, is accessible and appropriate in groups of different age, sex, ethnicity, education or income levels, and does not require expensive equipment. Walking is also known to reduce social barriers among people of different socio-economics status [57].

Contrary to other reports [27,28,31,51,52], the present study indicated the preferred exercise intensity as mild. Exercise guidelines for cancer patients suggest they should engage in at least moderate exercise [58]. Mild intensity could be the choice to start an exercise program, especially with physically ‘deconditioned’ people, and should be gradually increased to moderate and vigorous intensity. Several reviews show moderate-to-vigorous but not mild exercise intensity is effective in managing cancer side effects, and improves physical function [14,59]. 

In light of this evidence, the present findings highlight the need to inform cancer patients and their caregivers about the safety of moderate and vigorous intensities exercise. Patients’ exercise levels were related to their educational level, type of treatment and body fatness. Several studies have investigated the determinants and triggers of exercise behaviors in patients, but with inconsistent findings [60,61,62].

This appears to be the first study investigating the determinants of exercise preferences in Italian cancer patients before they were involved in exercise intervention programs. The study results provide useful data for planning future exercise programs. The self-reported QEX permitted the collection of a large amount of data and was quickly administered, without much burden on respondents, or costs. Another point of strength is the collection of information about why individuals did not wish to take part in the study. 

Limitations of the study need to be noted: the QEX information was self-reported and therefore open to several sources of bias. The QEX was filled and returned anonymously, so social desirability bias (for instance, patients may exaggerate their physical activity so as not to ‘disappoint’ the researcher) is less likely. The information leaflet given to patients at recruitment provides minimal information presenting the study but does not contain any recommendations/guidelines. However, just having provided information might have influenced the replies. Another potential source of error is selection bias: cancer patients who agreed to participate in the survey may be individuals more interested in exercise. To ensure a representative sample of patients, a random sample of outpatients was selected. Finally, the questionnaire does not serve to classify exercise adherence according to the new ACSM [58] guidelines for cancer patients. These guidelines were released in October 2019, after the QEX had been administered to the study sample of patients [58]. Nevertheless, the QEX classifies patients according to the previous ACSM guidelines [37]. This allows us to compare patients’ exercise levels with the studies that have been reported so far. Classification of the LSI according to the ACSM guidelines for cancer patients [37] allows a full comparison of study finding with the majority of other studies in the field. Nevertheless, this classification may have artificially inflated the percentage of participants who reported insufficient physical activity. The QEX does not collect information about participants’ pre-diagnosis exercise and physical activity and that limit its ability to explore associations with other possible determinants of current exercise behavior. The patients in STIP-ON were sampled to be representative of those attending the Verona oncology clinic (and not the full total of patients). Therefore, although more severe patients with severe comorbidities are likely to have been excluded, patients’ responses may also have been influenced by other comorbidities that were not investigated by the QEX.

Information from this survey is clinically relevant and may help in designing personalized interventions so cancer patients will achieve sufficient exercise/PA. Here are a few examples: (i) Since about 90% of participants said they wanted or needed a helper during the program, a targeted intervention program should include specific activities (and support) for helpers patients will nominate; (ii) Because about 30% of respondents said they prefer to exercise with other patients, exercise classes specifically for them and “learning from peers” social occasions should be organized; (iii) The majority of patients were insufficiently active and preferred mild exercise or slow walking. So as not to leave anyone behind, for those who are not able to engage in moderate exercise, a mild flexible entry program should be offered according the patient’s condition and preferences and then progress slowly towards higher-intensity exercise.

## 5. Conclusions

In conclusion, an exploratory survey like STIP-ON could serve as a necessary first step in developing lifestyle improvement interventions for patients. This is particularly important in a country like Italy where there is little knowledge in this field, and factors such as the family environment and social support are not well understood. Only a small proportion of patients were sufficiently active, although the majority were willing to start an exercise program. Exercise preferences in cancer patients tended to vary substantially. These findings underline the urgency of promoting personalized exercise intervention programs among Italian cancer patients.

## Figures and Tables

**Figure 1 ijerph-17-05351-f001:**
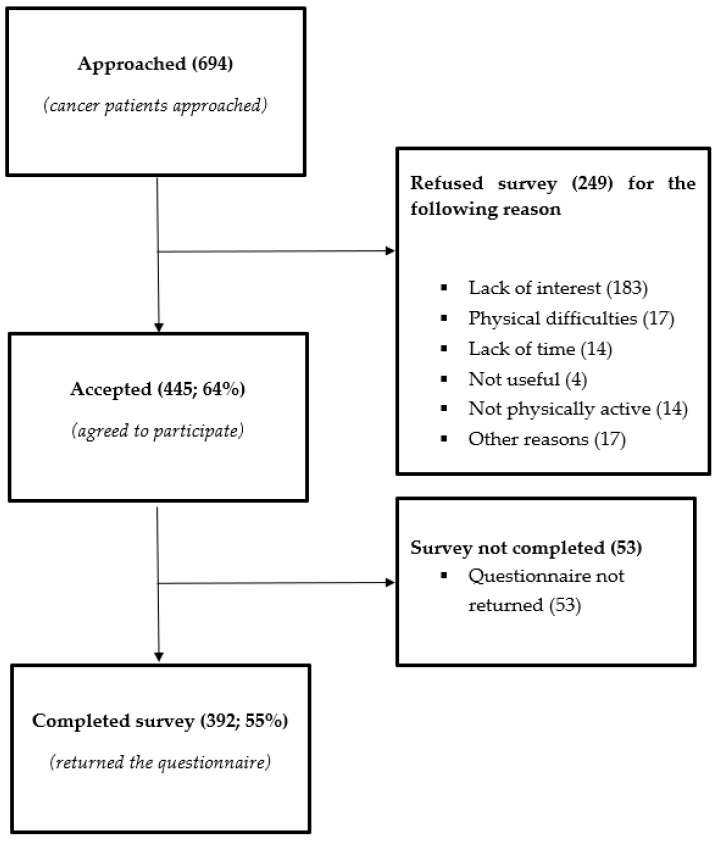
Flow of participants through the study.

**Table 1 ijerph-17-05351-t001:** General and tumor characteristics of 392 cancer patients^†^ according to willingness to participate in a specifically designed exercise program.

	All	Willingness to Participate ^‡^						*p*-Value ^§^
		Yes		Maybe		No		
	(392)	(179)		(134)		(79)		
	No.	%	No.	%	No.	%	No.	
**Age** (years)								
<65	239	48	115	38	91	14	33	0.023
≥65	149	42	63	28	41	30	45	
**Sex**								
Female	238	51	121	32	77	17	40	0.023
Male	154	38	58	37	57	25	39	
**Province of residence**								
Verona	244	43	105	34	82	23	57	0.114
Other	148	50	74	35	52	15	22	
**Education**								
Elementary (up to 10–11 years)	32	28	9	28	9	44	14	0.002
Secondary (up to 14 years)	119	41	49	37	44	22	26	0.451
Secondary (up to 18–19 years)	162	50	80	36	59	14	22	0.031
College/University	52	54	28	25	13	21	11	0.308
Postgraduate	18	50	9	33	6	17	3	0.920
**Body Mass Index ^¶^**								
Underweight	19	58	11	32	6	11	2	0.455
Normal weight	228	46	104	34	78	20	46	0.994
Overweight	107	41	44	36	39	22	24	0.503
Obese	30	57	17	27	8	17	5	0.461
**Marital status**								
Single	51	43	22	35	18	22	11	0.903
Married	269	45	122	35	94	20	53	0.819
Divorced	35	46	16	34	12	20	7	0.999
Widowed	34	56	19	24	8	21	7	0.371
**Occupational status**								
Retired	161	46	74	30	48	24	39	0.151
Homemaker	43	44	19	30	13	26	11	0.622
Part-time employed	45	49	22	42	19	9	4	0.118
Full-time employed	123	46	57	38	47	15	19	0.242
Other	20	35	7	35	7	30	6	0.462
**Perceived income adequacy ^††^**								
Inadequate	28	50	14	36	10	14	4	0.746
Barely adequate	120	45	54	40	48	15	18	0.152
Adequate	180	44	80	34	61	22	39	0.676
More than adequate	61	51	31	23	14	26	16	0.103
**Exercise level ^‡‡^**								
Insufficiently active	363	45	162	35	127	20	75	0.338
Active	27	59	16	26	7	15	4	
**Tumor site**								
Breast	101	54	55	31	31	15	15	0.096
Lung	22	41	9	36	8	23	5	0.894
Colorectum	39	31	12	49	19	21	8	0.091
Head/neck	9	44	4	22	2	33	3	0.554
Upper gastro-intestine	166	46	77	35	58	19	31	0.821
Gynecological	8	50	4	38	3	13	1	0.862
Urogenital	19	53	10	21	4	26	5	0.450
Melanoma	14	21	3	29	4	50	7	0.015
Other	14	36	5	36	5	29	4	0.659
**Disease status**								
Unknown	53	40	21	32	17	28	15	0.274
In remission/cured	62	56	35	27	17	16	10	0.178
Early	86	48	41	37	32	15	13	0.411
Advanced	85	38	32	40	34	22	19	0.239
Metastatic	106	47	50	32	34	21	22	0.866
**Treatments ^§§^**								
Surgery	215	44	95	32	69	24	51	0.167
Chemotherapy	329	44	144	36	119	20	66	0.102
Radiation therapy	113	44	50	29	33	27	30	0.119
Hormone therapy	50	56	28	30	15	14	7	0.249
Other	27	59	16	19	5	22	6	0.190
**Treatment status**								
About to start	11	55	6	27	3	18	2	0.829
Ongoing	325	44	144	35	115	20	66	0.452
Completed	35	51	18	29	10	20	7	0.728
Unknown	21	52	11	29	6	19	4	0.804
**Time from diagnosis**								
≤30 months	178	48	86	37	65	15	27	0.080
≥30 months	214	43	93	32	69	24	52	

^†^ Participants of STIP-ON study conducted in Verona, Italy, from July 2018 to April 2019. ^‡^ Willingness to participate in exercise program assessed by the question: Would you be interested in participating in an exercise program designed for cancer patients? ^§^ Pearson’s chi-squared used the null hypotheses of no association between physical exercise level and other patient/disease characteristics. ^¶^ Body Mass Index categories are those of the World Health Organization [34]. ^††^ Perceived income adequacy assessed by the question: Does your monthly income cover your monthly expenditure? ^‡‡^ Exercise level according to Leisure Score Index (LSI). Patients are active if LSI ≥24 and insufficiently active if LSI <24 [36]. ^§§^ Treatments, which may be completed or in course, and are not mutually exclusive.

**Table 2 ijerph-17-05351-t002:** Characteristics of preceding week’s exercise in cancer patients† by age and sex.

Exercise Frequency (Times/Week) by Intensity ^‡^	All Patients		Age < 65 Years				Age ≥ 65 Years			
	(392)		Men (81)		Women (158)		Men (71)		Women (78)	
	Mean (SD) ^§^	Median (IQR) ^¶^	Mean (SD) ^§^	Median (IQR) ^¶^	Mean (SD) ^§^	Median (IQR) ^¶^	Mean (SD) ^§^	Median (IQR) ^¶^	Mean (SD) ^§^	Median (IQR) ^¶^
Strenuous	0.20	0	0.21	0	0.29	0	0.13	0	0.08	0
	(0.84)	(0)	(0.85)	(0)	(1.03)	(0)	(0.70)	(0)	(0.42)	(0)
Moderate	0.71	0	0.64	0	0.77	0	0.61	0	0.75	0
	(1.43)	(1)	(1.30)	(1)	(1.40)	(1)	(1.53)	(0)	(1.53)	(1)
Mild	1.56	0	1.20	0	1.31	0	2.24	2	1.68	0
	(2.15)	(2)	(1.68)	(2)	(1.99)	(2)	(2.49)	(4)	(2.38)	(3)
**Exercise level ^††^**	N	%	N	%	N	%	N	%	N	%
Insufficiently active	363	93	76	94	144	92	66	94	74	95
Sufficiently active	27	7	5	6	13	8	4	6	4	5
**Sweat-inducing activity ^‡^**	N	%	N	%	N	%	N	%	N	%
Often	60	16	9	11	23	15	15	21	13	17
Sometimes	121	32	28	35	54	35	16	23	21	27
Rarely/never	204	53	43	54	77	50	39	56	43	56

^†^ Participants of STIP-ON study conducted in Verona, Italy, from July 2018 to April 2019. ^‡^ Exercise intensity according to [35]. ^§^ SD, standard deviation; ^¶^ IQR, interquartile range. ^††^ Exercise level according to Leisure Score Index (LSI): active ≥24; insufficiently active <24 [36].

**Table 3 ijerph-17-05351-t003:** Exercise preferences in cancer patients ^†^.

Preference as Expressed by Answers to Questions	%	No.
**Are you interested in participating in an exercise program designed for cancer patients? (392)**		
Yes	46	179
No	20	79
Maybe	34	134
**Who would you prefer to receive exercise instructions from? (392) ^‡^**		
Oncologist	57	224
Nurse	7	26
Kinesiologist	20	80
Nutritionist	20	80
Physiotherapist	30	118
Another cancer patient	3	11
No preference	20	79
Other	2	8
**How would you prefer to receive exercise instructions? (376)**		
Face to face	72	270
By telephone	3	13
Videotape	2	9
Television	1	3
Leaflet/pamphlet	5	20
Over the internet	3	13
No preference	12	46
Other	1	2
**Where would you prefer to exercise? (378)**		
At home	21	78
At a community fitness center	12	44
At an adapted exercise fitness center	22	83
Outside	27	103
No preference	18	70
Other	1	2
**What time of day would you prefer to exercise? (376)**		
Morning	48	179
Afternoon	31	118
Evening	9	32
No preference	13	48
**In what part of the week would you prefer to exercise? (367)**		
Weekday	70	256
Weekend	9	32
No preference	22	79
**How would you prefer to exercise? (363)**		
Unsupervised	15	56
Supervised	62	224
No preference	23	83
**What kind of exercise program would you prefer? (360)**		
Individual with a program to follow at home	27	96
Individual with personal trainer	25	90
In a group with a kinesiologist/physiotherapist/exercise specialist	40	144
Other	8	30
**Would you like session content to vary? (363)**		
Same each time	29	105
Different each time	34	123
No preference	37	135
**Who would you prefer to exercise with? (373)**		
Nobody	16	61
Other cancer patients	27	104
Family members	8	29
Friends	8	30
A group	13	47
No preference	27	101
Other	1.3	5
**Who would you want as “helper” during the program? (369)**		
Nobody	13	48
Exercise specialist	22	83
Neighbor	1	5
Colleague	1	3
Friend	19	71
Son/daughter	13	47
Spouse	28	102
Other relative	3	10
**How often would you prefer to exercise? (365)**		
Never	1	5
Once a week	15	54
Twice a week	37	136
Three times a week	30	111
Four times a week	5	19
Five times a week	5	19
Six times a week	1	5
Seven times a week	4	16
**What exercise intensity would you prefer? (376)**		
Mild	48	175
Moderate	39	141
Strenuous	7	24
No preference	6	23

^†^ Participants of STIP-ON study conducted in Verona, Italy, from July 2018 to April 2019 ^‡^ Replies add up to more than 694 as participants could choose more than one instructor.

**Table 4 ijerph-17-05351-t004:** Multivariable logistic modeling of associations of characteristics of cancer patients ^†^ with willingness ^‡^ to participate in an exercise program.

		All (No.)	Willing to Participate (No.)	Minimally-Adjusted Model ^§^			Fully-Adjusted Model ^¶^		
				OR ^††^	95% CI^††^	*p*-Value ^††^	OR ^††^	95% CI^††^	*p*-Value ^††^
Age	≤65 y (reference)	239	115	1			1		
	≥65y	149	63	0.84	0.55; 1.28	0.424	0.63	0.34; 1.15	0.085
Sex	Women (reference)	238	121	1			1		
	Men	154	58	0.61	0.40; 0.93	0.021	0.55	0.32; 0.94	0.029
Education	Up to age 14 years (reference)	151	58	1			1		
	Beyond age 14 years	241	121	1.60	1.04; 2.46	0.031	1.87	1.15; 3.04	0.011
Residence	Outside city (Reference)	148	74	1			1		
	In Verona	244	105	0.70	0.46; 1.07	0.100	0.61	0.38; 0.99	0.045
Perceived income adequacy	Inadequate (reference)	148	68	1			1		
	Adequate	244	111	1.06	0.69; 1.61	0.764	0.94	0.58; 1.51	0.785
Marital status	Married (reference)	269	122	1			1		
	Single	51	22	0.77	0.41; 1.44	0.366	0.64	0.31; 1.31	0.213
	Divorced	35	16	0.92	0.45; 1.89		0.94	0.43; 2.01	
	Widowed	34	19	1.88	0.85; 4.16		2.51	1.04; 6.04	
Occupational status	Retired (reference)	161	74	1			1		
	Homemaker	43	19	0.66	0.31; 1.33	0.354	0.73	0.33; 1.61	0.203
	Part-time employed	45	22	0.83	0.37; 1.62		0.67	0.30; 1.50	
	Full-time employed	123	57	0.83	0.46; 1.43		0.68	0.36; 1.29	
	Other	20	7	0.52	0.17; 1.33		0.58	0.18; 1.90	
Frequency of sweat-inducing activity	<1 time/week (reference)	204	84	1			1		
	1–2 times/week	121	61	1.46	0.92; 2.32	0.037	1.50	0.91; 2.25	0.035
	>2 times/week	60	33	1.79	1.00; 3.23		1.92	1.92; 3.63	
Tumor site ^§§,¶¶^	Breast (reference)	101	55	1			1		
	Lung	22	9	0.60	0.24; 1.58	0.251	0.52	0.16; 1.68	0.423
	Colorectal	39	12	0.39	0.18; 0.86		0.46	0.18; 1.16	
	Upper gastro-intestine	166	77	0.73	0.44; 1.20		0.62	0.31; 1.24	
	Urogenital system	19	10	1.00	0.37; 2.75		1.02	0.28; 3.67	
	Melanoma	14	3	0.23	0.06; 0.88		0.13	0.03; 0.64	
	Other sites^6^	31	13	0.66	0.29; 1.51		0.52	0.20; 1.40	
Disease status	Remission (reference)	62	35	1			1		
	Early	86	41	0.70	0.36; 1.36	0.145	0.61	0.27; 1.37	0.595
	Advanced	85	32	0.48	0.25; 0.95		0.65	0.29; 1.44	
	Metastatic	106	50	0.71	0.37; 1.44		0.72	0.30; 1.73	
	Unknown	53	21	0.51	0.24; 1.08		0.83	0.33; 2.06	
Chemotherapy	No (reference)	55	31	1			1		
	Yes	329	144	0.51	0.28; 0.92	0.026	0.45	0.23; 0.86	0.016
Surgery	No (reference)	169	80	1			1		
	Yes	215	95	0.91	0.60; 1.37	0.644	1.07	0.67; 1.71	0.787
Radiotherapy	No (reference)	271	125						
	Yes	113	50	0.90	0.57; 1.41	0.650	0.97	0.58; 1.61	0.900
Hormone therapy	No (reference)	334	147						
	Yes	50	28	1.45	0.78; 2.71	0.239	1.66	0.83; 3.32	0.152
Other treatments	No (reference)	365	163						
	Yes	27	16	2.10	0.92; 4.82	0.077	1.89	0.81; 4.39	0.142
Treatment status	Completed (reference)	36	18	1			1		
	About to start	11	6	1.46	0.36; 5.84	0.695	1.80	0.31; 10.5	0.781
	Ongoing	325	144	0.79	0.37; 1.60		0.75	0.33; 1.71	
	Unknown	21	11	1.27	0.42; 3.83		1.72	0.45; 6.48	
Time from diagnosis	≤30 months (reference)	178	86	1			1		
	>30 months	214	93	0.79	0.53; 1.19	0.265	0.68	0.42; 1.13	0.126

^†^ Participants of STIP-ON study conducted in Verona, Italy, from July 2018 to April 2019 ^‡^ Willingness classified as yes vs. no/maybe. ^§^ Age- and sex-adjusted models, unless otherwise specified. ^¶^ Each variable adjusted for the following, unless otherwise specified: Sex (man vs. woman); Age (<65 y vs. ≥65 y); Education (more than 14 years of age vs. up to 14 years); Residence (outside city vs. win city of Verona); Perceived income adequacy (adequate vs. inadequate); Marital status (married, single, divorced, widowed); Occupational status (retired, homemaker, part-time employed, full-time employed, other); Frequency of sweat-inducing activity (<1 time/week, 1–2 times/week, >2 times/week); Tumor site (breast, lung, colorectum, upper gastro-intestine, urogenital system, melanoma, other); Disease status (in remission, early, advanced, metastatic, unknown); Chemotherapy (yes vs. no); Surgery (yes vs. no); Radiotherapy (yes vs. no); Hormone therapy (yes vs. no); Other treatments (yes vs. no); Treatment status (Completed, About to start, Ongoing, Unknown); Time from diagnosis (≤30 months, >30 months); “Lack of preference” variable (score 0 for no no-preference reply, score 1 for 1 no-preference reply, score 2 for 2 or more no-preference replies given to exercise preference questions); “Independence” variable (score 0 if “on my own” never chosen in exercise preference questions, score1 otherwise. ^††^ OR (odds ratios), CI (confidence intervals), and *p*-values from multivariable logistic regression model. ^§§^ Tumor sites with less than 10 patients are classified as “other site.” ^¶¶^ Models investigating tumor site (both minimally and fully adjusted models) were not adjusted for sex.

**Table 5 ijerph-17-05351-t005:** Multivariable regression modeling of associations of characteristics of 392 cancer patients ^†^ with exercise level ^‡^.

		No.	Exercise Level ^†^		Minimally-Adjusted Model ^§^			Fully-Adjusted Model ^¶^		
			Mean	SD	β ^††^	95% CI ^††^	*p* Value ^††^	β ^††^	95% CI ^††^	*p* Value ^††^
Age	<65 y (reference)	238	0.61	1.14	Ref			Ref		
	≥65y	148	0.42	0.91	−0.17	−0.39; 0.05	0.125	0.03	−0.24; 0.31	0.801
Sex	Female (reference)	237	0.58	1.10	Ref			Ref		
	Male	153	0.46	0.99	−0.10	−0.32; 0.12	0.378	−0.08	−0.32; 0.16	0.489
Body Mass Index (kg/m^2^)	<25 (reference)	245	0.67	1.21	Ref			Ref		
	≥25	137	0.31	0.71	−0.34	−0.57; −0.11	0.003	−0.33	−0.57; −0.10	0.005
Education	Up to age 14 years (reference)	149	0.32	0.75	Ref			Ref		
	Beyond age 14 years	241	0.67	1.19	0.32	0.10; 0.55	0.004	0.32	0.09; 0.55	0.005
Perceived income adequacy	Inadequate (reference)	147	0.52	1.14	Ref			Ref		
	Adequate	243	0.55	1.01	0.06	−0.16; 0.28	0.581	0.03	−0.20; 0.25	0.826
Marital status	Married (reference)	275	0.53	1.00	Ref			Ref		
	Single/other	51	0.63	1.15	0.04	−0.28; 0.37	0.86	−0.05	−0.39; 0.28	0.781
	Divorced	35	0.63	1.52	0.08	−0.30; 0.46		0.01	−0.37; 0.39	
	Widowed	34	0.41	0.90	−0.06	−0.46; 0.34		−0.07	−0.47; 0.34	
Occupation	Retired (reference)	160	0.38	0.83	Ref			Ref		
	Homemaker	43	0.40	0.87	−0.05	−0.42; 0.33	0.048	0.02	−0.37; 0.41	0.073
	Part-time employed	45	0.66	1.14	0.19	−0.14; 0.62		0.24	−0.16; 0.64	
	Full-time employed	123	0.76	1.34	0.37	0.07; 0.66		0.32	−0.01; 0.62	
	Other	19	0.38	0.74	−0.01	−0.53; 0.51		0.09	−0.48; 0.66	
Tumor site ^‡‡^	Breast (reference)	101	0.63	1.27	Ref			Ref		
	Lung	22	0.09	0.33	−0.49	−0.97; 0.02	0.579	−0.52	−1.05; 0.01	0.926
	Colorectum	39	0.32	0.94	−0.30	−0.67; 0.12		−0.33	−0.76; 0.10	
	Upper gastro-intestine	164	0.59	1.02	−0.02	−0.29; 0.24		−0.03	−0.36; 0.31	
	Urogenital system	19	0.39	0.94	−0.18	−0.70; 0.35		−0.26	−0.82; 0.31	
	Melanoma	14	0.79	1.10	0.17	−0.43; 0.76		0.04	−0.61; 0.68	
	Other site ^§§^	31	0.53	0.92	−0.15	−0.51; 0.34		−0.18	−0.65; 0.30	
Disease status	Remission (reference)	62	0.85	1.25	Ref			Ref		
	Early	85	0.73	1.31	−0.10	−0.44; 0.25	0.006	0.15	−0.24; 0.54	0.010
	Advanced	84	0.30	0.82	−0.51	−0.86; −0.15		−0.36	−0.75; −0.02	
	Metastatic	106	0.50	0.97	−0.30	−0.63; 0.03		−0.28	−0.59; 0.08	
	Unknown	53	0.31	0.69	−0.49	−0.87; −0.10		−0.43	−0.81; 0.03	
Chemotherapy	No (reference)	55	0.56	1.05	Ref			Ref		
	Yes	327	0.54	1.07	−0.06	0.36; 0.25	0.720	0.04	−0.34; 0.38	0.914
Surgery	No (reference)	167	0.48	0.98	Ref			Ref		
	Yes	215	0.58	1.12	0.10	−0.12; 0.31	0.366	0.07	−0.16; 0.31	0.540
Radiotherapy	No (reference)	269	0.55	1.10	Ref			Ref		
	Yes	113	0.50	0.97	−0.03	−0.26; 0.21	0.833	−0.10	−0.36; 0.16	0.454
Hormone therapy	No (reference)	332	0.53	1.07	Ref			Ref		
	Yes	50	0.58	1.00	−0.01	−0.34; 0.32	0.955	0.10	−0.26; 0.46	0.581
Other treatments	No (reference)	363	0.53	1.05	Ref			Ref		
	Yes	27	0.60	1.10	0.09	−0.33; 0.51	0.682	0.10	−0.37; 0.57	0.684
Treatment status	Completed (reference)	35	0.83	1.31	Ref			Ref		
	About to start	11	0.54	1.62	−0.20	−0.93; 0.53	0.184	0.18	−0.67; 1.04	0.462
	Ongoing	323	0.50	1.00	−0.30	−0.67; 0.07		−0.14	−0.54; 0.26	
	Unknown	21	0.54	1.08	−0.21	−0.79; 0.37		−0.08	−0.72; 0.55	
Time from diagnosis	≤30 months (reference)	177	0.47	0.94	Ref			Ref		
	>30 months	213	0.59	1.15	0.13	−0.08; 0.34	0.225	0.15	−0.08; 0.39	0.207

^†^ Participants of STIP-ON study conducted in Verona, Italy, from July 2018 to April 2019 ^‡^ Exercise level assessed using Leisure Score Index [36]. ^§^ Age and sex adjusted models unless otherwise specified. ^¶^ Each variable was adjusted for the following, unless otherwise specified: Sex (man vs. woman); Age (<65y vs. ≥65y); Education (beyond 14 years of age vs. up to 14 years); Residence (outside city vs. in city of Verona); Perceived income adequacy (adequate vs. inadequate); Marital status (married, single, divorced, widow); Occupational status (retired, homemaker, part-time employed, full-time employed, other); Tumor site (breast, lung, colorectum, upper gastro-intestine, urogenital system, melanoma, other); Disease status (remission, early, advanced, metastatic, unknown); Chemotherapy (yes vs. no); Surgery (yes vs. no); Radiotherapy (yes vs. no); Hormone therapy (yes vs. no); Other treatments (yes vs. no); Treatment status (completed, about to start, ongoing, unknown); Time from diagnosis (≤30 months vs. 30 months). ^††^ Beta coefficients 9 (β), confidence intervals (CI), and *p* values from multivariable regression models. The β coefficient is the amount of change in exercise level (Leisure Score Index) in each category of predictor variable compared to reference. ^‡‡^ Minimally and fully adjusted models for tumor site not adjusted for sex. ^§§^ Tumor sites with <10 patients classified with another site.

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
