# Peer review of "Exercise Levels and Preferences in Cancer Patients: A Cross-Sectional Study"

_ijerph, 2020, doi:10.3390/ijerph17155351_

Round 1

Reviewer 1 Report

This is a well written and clearly presented manuscript which comprehensively articulates the exercise preferences among cancer patients. 

Strengths of this study are its relevance among the cancer population for understanding physical activity and exercise behaviours to optimise interventions targeting increased physical activity engagement and improved quality of life, health and wellbeing markers and side-effects.

As a researcher in physical activity and sedentary behaviour I found myself questioning why there was a specific focus on exercise, rather than, more broadly, physical activity which has also demonstrated numerous benefits to physical and mental health outcomes for healthy populations and people with cancer. Particularly so, as many of the preferred strategies over summer or winter were identified as walking activities. I feel this categorisation may have artificially inflated the percentage of participants who reported insufficient activity? I would have been interested to know about participants pre-diagnosis exercise and physical activity engagement as this was omitted. I see you have shown an association with education, but perhaps a missed opportunity to explore associations with habitual physical activity as well.  

Below I have identified some areas which I think could be improved/ clarified with some minor corrections. I have presented this by section and where possible, provided a line reference number (line number taken from PDF version not word). 

Minor revisions:

Introduction

Line 35- I think your opening two sentences could be restructured to improve clarity as it currently reads as a bit vague i.e. man years ago.

Line 36- I think it would be useful for the reader to clarify what is meant by ‘survival’ rate. In the literature survival rate is usually discussed as >5 years from diagnosis.

Line 41-42 Can you clarify whether you are referring to an active lifestyle in general, or specifically for cancer patients.

Line 51. Can you include any statistics to support this statement? You have provided this in your discussion but I think it would be useful to include here in the introduction.

Line 53. This suggests that the only barriers to physical activity for healthy populations are lack of time, costs and logistics- when in fact these are a few examples of a multitude of factors which influence participation in physical activity.

Line 55. Please clarify whether you are referring to engagement among healthy populations or cancer patients.

Line 56-65. I think you could strengthen the rationale and emphasise the originality of your research here.  

 Methods

Line 69- Please clarify why this is a representative sample of patients i.e which characteristics are representative?

Line 76- Please clarify the term in loco

Line 75. You mention a brochure here. I would be interested to know what information was included for the participants in the brochure. Was this informed by any literature/ guidelines? Were they informed about the physical activity recommendations before completing the questionnaire? If so this may introduce some potential risk for bias. Would it be possible to include this brochure as a supplementary  material?

Line 78- Please clarify- if the survey was anonymous, how did you check for duplication? Were there any incidences of participants answering the questionnaire twice?

Line 79- You state past cancer diagnosis is an inclusion criteria but you include currently diagnosed participants also.

Line 92-94- You report accurately regarding the number of declines to take part and reasons for doing so which is excellent, and really useful information. I was wondering here, how many took the survey home? You report 53 did not return the survey to the centre.

Line 110- I think there is a missed opportunity to capture non-exercise physical activity here i.e. gardening and household chores are considered active and beneficial for health. The International Physical Activity Questionnaire (IPAQ) survey is able to capture this data which breaks activity into domain specific areas and also asks for the frequency, intensity and duration of say activity. Your survey only captures frequency and intensity, which I think should be acknowledged within the limitations section. It will not be possible to repeat this among the sample of participants but I wondered if you could provide a rationale for use of this measure over alternative self-report measures which are valid and reliable.

Results

As I mentioned previously, it is really useful that you have the data on how many people were approached and the reasons for declining to take part. Just a note to perhaps also include the number of people who completed the survey from home.

In table 4 you present education and association with willingness to participate in exercise. Again I would have been interested to know who was a habitual exerciser pre-diagnosis- not sure whether you will have that data available?

Discussion

General comments regarding the discussion; you appear to have some very short sections of text which do not conform to a typical paragraph format. Thus they appear to be stand-alone statements which could be better integrated within the surrounding text.

Line. 296. I think it would be useful for the reader to provide some examples of the characteristics you are referring to here. Also, your reference to Vallance et al and Karvinen is not overly clear to the reader so I would suggest considering the sentence structure here.

Line 308. Similarly, your statement about patient preferences regarding exercise instructions the current standard of care should be merged with the following sentence to make the point clearer.

Line 338. Is there a specific recommendation for the time spent in moderate exercise?

Line 349. ‘This is important because it emphasizes that no one wants (and therefore should not) 349 be left behind at any stage of the disease’ seems to be a broad statement and I feel this could be better linked back to your aim/ physical exercise.

Line 356-358. Check sentence structure ‘ This is a new finding, particularly relevant as it underlines how the psychological status, as well as the physical limits, must be taken 357 into account when encouraging people to improve their exercise level’

Line 363- 366. Your discussion of social desirability appears to contradict itself here. Firstly it is not clear why social desirability may be a limitation, but then you state that an anonomous survey would mitigate this risk of bias. As I mentioned previously, I think you overlook some of the methodological limitations of the use of the survey. Also a strength of the data collected includes your insight into why individuals did not wish to take part.

Line 367. Check sentence structure ‘ Another possible systematic is the selection bias

Line 374 I think the ACSM guidelines need to appear earlier in your paper either in the introduction or methods section where you describe the LSI index <24> meaning insufficient activity so it would be important for the reader if you acknowledge which ACSM guidelines you are basing this on sooner.

Author Response

Point-by-point reply reviewer 1

This is a well written and clearly presented manuscript which comprehensively articulates the exercise preferences among cancer patients. 

Strengths of this study are its relevance among the cancer population for understanding physical activity and exercise behaviours to optimise interventions targeting increased physical activity engagement and improved quality of life, health and wellbeing markers and side-effects.

  1. a) As a researcher in physical activity and sedentary behaviour I found myself questioning why there was a specific focus on exercise, rather than, more broadly, physical activity which has also demonstrated numerous benefits to physical and mental health outcomes for healthy populations and people with cancer. Particularly so, as many of the preferred strategies over summer or winter were identified as walking activities. I feel this categorisation may have artificially inflated the percentage of participants who reported insufficient activity?

Reply a) : Thank you for this important observation and for giving us the opportunity to clarify this point. It is true that physical activity - not only ‘exercise’ - plays an important role in improving the quality of life for cancer patients and survivors. However, the ACSM guidelines for oncological patients (both 2010 and 2019  release) recommend raising the level of exercise (and not the more generic ‘level of physical activity’). The meta-analysis by Buffart and colleagues (REF #56 Buffart LM,. Cancer Treat Rev. 2017) shows that the maximum benefit in terms of quality of life and physical function in patients with cancer is achieved with  supervised exercise interventions. We are currently planning an intervention study on cancer patients to improve exercise levels in practice. Thus, to implement our next step better it was important to understand the exercise behavior of this population. We recognize that leisure time activities are underestimated in our study  and have included this among the limits of the study.

We have added the following in the Discussion

Classification of the LSI according to the ACSM guidelines for cancer patients [37] allows a full comparison of study finding with the majority of other studies in the field. Nevertheless, this classification may have artificially inflated the percentage of participants who reported insufficient physical activity.

  1. b) I would have been interested to know about participants’ pre-diagnosis exercise and physical activity engagement as this was omitted. I see you have shown an association with education, but perhaps a missed opportunity to explore associations with habitual physical activity as well.

Reply b): Thanks for the comment. This information was not collected by the QEX; the main reason was to limit the number of QEX items so as not to overburden patients (20 minutes maximum for compilation). We have now noted the absence of this information among the limits of the study and have added the following in the Discussion:

The QEX does not collect information about participants’ pre-diagnosis exercise and physical activity and that limit its ability to explore associations with other possible determinants of current exercise behavior.

Minor revisions:

Introduction

Line 35- I think your opening two sentences could be restructured to improve clarity as it currently reads as a bit vague, eg ‘many years ago’.

Reply: the sentences have been rephrased as follows:

In 2019  it was estimated that about 3.5 million Italians (5.3% of the entire population) are living after a cancer diagnosis [1]. Improvements in medical treatments have led to a substantial increase in the proportion of cancer patients with death rates similar to those of the general Italian population [2].

Line 36- I think it would be useful for the reader to clarify what is meant by 'survival' rate. In the literature survival rate is usually discussed as >5 years from diagnosis.

Reply: To avoid confusion we have rephrased the sentence, using more precise terms to define “…Italians living after a cancer diagnosis" and "proportion of patients with death rates similar to those of the general Italian population"   

Line 41-42 Can you clarify whether you are referring to an active lifestyle in general, or specifically for cancer patients.

Reply: We refer to an active lifestyle specifically for cancer patients. The sentence has been rephrased as follows:

There is growing evidence that in cancer patients (especially breast, colon and prostate) [5-7] an active lifestyle is associated with a lower risk of recurrence and mortality. 

Line 51. Can you include any statistics to support this statement? You have provided this in your discussion but I think it would be useful to include here in the introduction.

Reply: We have changed the sentence as follows: “...

Despite the benefits related to PA and exercise a large percentage of cancer patients from 25% to 84% are not sufficiently active [17-19] and the level of exercise has been seen to decrease after cancer diagnosis [20].

Line 53. This suggests that the only barriers to physical activity for healthy populations are lack of time, costs and logistics when in fact these are a few examples of a multitude of factors which influence participation in physical activity.

Reply: We have changed the sentence as follows:

A multitude of factors influence the participation  of  the general population  in  exercise programs (e.g. lack of time, cost, logistic difficulties, etc.) [21]. Cancer patients face further obstacles on account of their condition (e.g. cancer-related fatigue, muscle weakness, nausea, sleep disorders) [22, 23].

Line 55. Please clarify whether you are referring to engagement among healthy populations or cancer patients.

Reply: We have modified the text as follows:

To develop a successful exercise intervention, cancer patients’ barriers and preferences must be considered, allowing them to pick the activities they perceive as beneficial and enjoyable [23-26]. 

Line 56-65. I think you could strengthen the rationale and emphasise the originality of your research here.

Reply: we have rephrased the sentence, explaining the rationale and the originality of the study, as follows:

The rationale of the study is that understanding patients’ preferences and barriers to physical activity will make it easier for them to participate successfully in a future intervention study to improve their physical fitness.

Methods

Line 69- Please clarify why this is a representative sample of patients i.e which characteristics are representative?

Reply: Our sample is randomly selected among patients visiting the outpatients oncological department of the Borgo Roma Hospital in Verona.  This is an important oncological department  in Verona but it is not the only cancer hospital in the Verona province, so a random sample of its outpatients is not in fact a representative sample of the whole cancer population of the province.  However, the purpose of the study is to describe  the patient population to whom the future intervention will be addressed, so this sample is perfect  for  our purposes. This is now explained in the metods:

The STIP-ON sample was thought to be representative of patients visiting the cancer outpatients' facilities: on randomly selected days they were approached face to face, informed about the study and asked whether they would be willing to complete the questionnaire anonymously to investigate their characteristics and preferences regarding exercise. 

Line 76- Please clarify the term in loco

In loco stands for "on the spot" (rather than taking the QEX away to complete it at home). To avoid confusion we have replaced it with ‘on the spot’.

Line 75. You mention a brochure here. I would be interested to know what information the brochure gave participants. Was this based on any particular any literature/guidelines? Were they informed about the physical activity recommendations before completing the questionnaire? If so, this might introduce some potential risk of bias. Would it be possible to include this brochure as a supplementary material?

Reply: The brochure gives minimal information presenting the study but does not contain any recommendations/guidelines. However, we do realise that having provided information might have influenced the replies. As regards social acceptability bias  a distortion of the information provided could be caused because the interviewed person  tries to comply with the aims of the study itself (e.g.  patients could exaggerate their physical activity in order not to ‘disappoint’ the researcher) . This risk is inevitable since informing the patient about the aims and methods of the study is an obligation of the researcher. However, we believe that offering an anonymous survey minimized this bias. We have now included the informative brochure as supplementary material and have explained it in the Discussion by adding the following:

The QEX was filled and returned anonymously, so social desirability bias (for instance, patients may exaggerate their physical activity so as not to ‘disappoint’ the researcher) is less likely. The information leaflet given to patients at recruitment provides minimal information presenting the study but does not contain any recommendations/ guidelines.  However, just having provided information might have influenced the replies.

Line 78- Please clarify- if the survey was anonymous, how did you check for duplication? Were there any incidences of participants answering the questionnaire twice?

Reply: people were systematically asked by the staff if they had already completed the survey another time/day before QEX was administered. A duplicate check was also done, looking for duplicates by date of birth, province of residence, sex, education, anthropometry and marital status. No duplicate lines were found using this procedure. After reviewer 1’s  comment, just to make sure, we checked again for duplicates in a less conservative way without considering weight and height. So we restricted duplicated variables to the date of birth, province of residence, sex, education, occupation and marital status. Using this less conservative procedure we found 13 likely duplicates of the QEX, that were eliminated (the second QEX was dropped). This small decrease in the sample (from 707 to 694) obliged us to reformulate all the tables because even though the results and conclusions are unchanged there were small differences in the numbers. This is now explained in the methods where the following has been added:

Invited participants were systematically asked by the staff if they had already completed the survey another time/day before this QEX was administered.  A duplicate check was done, looking for duplicates by date of birth, province of residence, sex, education, and marital status.

Line 79- You state past cancer diagnosis is an inclusion criteria but you include currently diagnosed participants also.

Reply: We apologize for this misunderstanding: the sample included all kinds of  cancer survivors (incl. those whose diagnosis had just been made or was being defined). We have clarified this point in methods:.

Cancer patients' eligibility criteria were: age ≥18 years, a cancer diagnosis and adequate Italian language proficiency to answer the survey questionnaire (QEX). Invited participants included all kinds of cancer survivors (including those whose diagnosis had just been made or was being defined).

Line 92-94- You report accurately regarding the number of declines to take part and reasons for doing so which is excellent, and really useful information. I was wondering here, how many took the survey home? You report 53 did not return the survey to the centre.

Reply: Unfortunately we are unable to answer this question. We left the patients free to complete and return the QEX the same day or to take it home, fill it in and bring it back another day. Regarding the 53 people who did not return the QEX in the box, we do not  know whether they are classifiable as passive/involuntary drop-outs (eg, they accidentally forgot or lost the QEX) or if theirs was active drop-out (eg, they were annoyed by the questions and they “actively” did not return it, on purpose)

Line 110- I think there is a missed opportunity to capture non-exercise physical activity here i.e. gardening and household chores are considered active and beneficial for health. The International Physical Activity Questionnaire (IPAQ) survey is able to capture this data which breaks activity into domain specific areas and also asks for the frequency, intensity and duration of say activity. Your survey only captures frequency and intensity, which I think should be acknowledged within the limitations section. It will not be possible to repeat this among the sample of participants but I wondered if you could provide a rationale for use of this measure over alternative self-report measures which are valid and reliable.

Reply: Thank you for this important observation and for giving us the opportunity to clarify this point. It is true that physical activity - not only ‘exercise’ - plays an important role in improving the quality of life for cancer patients and survivors. However, the ACSM guidelines for oncological patients (both 2010 and 2019  release) recommend raising the level of exercise (and not the more generic ‘level of physical activity’). The meta-analysis by Buffart and colleagues (REF #56 Buffart LM,. Cancer Treat Rev. 2017) shows that the maximum benefit in terms of quality of life and physical function in patients with cancer is achieved with  supervised exercise interventions. We are currently planning an intervention study on cancer patients to improve exercise levels in practice. Thus, to implement our next step better it was important to understand the exercise behavior of this population. We recognize that leisure time activities are underestimated in our study  and have included this among the limits of the study.

We have added the following in the Discussion

Classification of the LSI according to the ACSM guidelines for cancer patients [37] allows a full comparison of study finding with the majority of other studies in the field. Nevertheless, this classification may have artificially inflated the percentage of participants who reported insufficient physical activity.

Results

As I mentioned previously, it is really useful that you have the data on how many people were approached and the reasons for declining to take part. Just a note to perhaps also include the number of people who completed the survey from home.

Reply: Unfortunately we are unable to answer this question. We left the patients free to complete and return the QEX the same day or to take it home, fill it in and bring it back another day. Regarding the 53 people who did not return the QEX in the box, we do not  know whether they are classifiable as passive/involuntary drop-outs  (eg, they accidentally forgot or lost the QEX) or if theirs was active drop-out (eg, they were annoyed by the questions and they “actively” did not return it, on purpose)

In table 4 you present education and association with willingness to participate in exercise. Again I would have been interested to know who was a habitual exerciser pre-diagnosis- not sure whether you will have that data available?

Thanks for the comment. This information was not collected by the QEX; the main reason was to limit the number of QEX items so as not to overburden patients (20 minutes maximum for compilation). We have now noted the absence of this information among the limits of the study and have added the following in the Discussion:

The QEX does not collect information about participants’ pre-diagnosis exercise and physical activity and that limit its ability to explore associations with other possible determinants of current exercise behavior.

Discussion

General comments regarding the discussion; you appear to have some very short sections of text which do not conform to a typical paragraph format. Thus they appear to be stand-alone statements which could be better integrated within the surrounding text.

Reply: Following this and a similar comment from reviewer 2, the discussion has now been completely re-structured. The main results are now put in relation to the literature and a tentative explanation of the results/possible mechanism is provided, where appropriate.

Line. 296. A) I think it would be useful for the reader to provide some examples of the characteristics you are referring to here.

  1. A) As stated above, the Discussion has been re-structured, and lines 296… now read as follows:

The STIP-ON survey found that only 7% of cancer patients do enough physical exercise. Previous studies reported the percentage of cancer patients with adequate exercise levels, between 16-85% [17].

Line. 296. B)  Also, your reference to Vallance et al and Karvinen is not overly clear to the reader so I would suggest considering the sentence structure here.

  1. B) Comparison of our results with literature is now explained better, and reads as follows:

Roughly 80% of patients were willing to start an exercise program designed for cancer patients. Previous studies reported similar results, finding that the majority of bladder [42], non-Hodgkin’s lymphoma [29], prostate [27], head and neck [31], endometrial [28],  ovarian [30] and breast [27] cancer survivors were interested in an exercise program.

Line 308. Similarly, your statement about patient preferences regarding exercise instructions the current standard of care should be merged with the following sentence to make the point clearer.

Reply: As stated above, the Discussion has been re-structured. The comment at (former)  line 308 now reads .....

Regarding the preferred source of exercise instruction, the oncologist was the preferred person to deliver instructions in the present survey and this is not  in line with the current standard of care.

Line 338. Is there a specific recommendation for the time spent in moderate exercise?

Yes, this is now mentioned in methods.

The ACSM guidelines (release 2010) suggest cancer patients engage in at least 150 minutes/week of moderate or 75 minutes/week of vigorous exercise [37].

Line 349. 'This is important because it emphasizes that no one wants (and therefore should not) to be left behind at any stage of the disease' seems to be a broad statement and I feel this could be better linked back to your aim/ physical exercise.

Thank you for the suggestion. We have rephrased the study rationale  as follows

The rationale of the study is that understanding patients’ preferences and barriers to physical activity will make it easier for them to participate successfully in a future intervention study to improve their physical fitness.

Line 356-358. Check sentence structure 'This is a new finding, particularly relevant as it underlines how the psychological status, as well as the physical limits, must be taken  into account when encouraging people to improve their exercise level.

Reply: thank you for pointing this out. We have dropped the sentence.

Line 363- 366. A) Your discussion of social desirability appears to contradict itself here. Firstly it is not clear why social desirability may be a limitation, but then you state that an anonymous survey would mitigate this risk of bias. As I mentioned previously, I think you overlook some of the methodological limitations of the use of the survey.

Reply: We do realise that having provided information might have influenced the replies. As regards social acceptability bias a distortion of the information provided could be caused because the interviewed person tries to comply with the aims of the study itself (e.g. patients could exaggerate their physical activity in order not to ‘disappoint’ the researcher). This risk is inevitable since informing the patient about the aims and methods of the study is an obligation of the researcher. However, we believe that offering an anonymous survey minimized this bias. We have now explained it in the Discussion by adding the following:

The QEX was filled and returned anonymously, so social desirability bias (for instance, patients may exaggerate their physical activity so as not to ‘disappoint’ the researcher) is less likely. The information leaflet given to patients at recruitment provides minimal information presenting the study but does not contain any recommendations/ guidelines.  However, just having provided information might have influenced the replies.

Line 363- 366. B) Also a strength of the data collected includes your insight into why individuals did not wish to take part.

  1. B) Thanks for the suggestion, we have added among the strengths of the study the collection of information about why individuals did not wish to take part.

Line 367. Check sentence structure 'Another possible systematic is the selection bias'

Reply: Thank you for pointing this out. We have rephrased it as follows:

Another potential source of error is selection bias:

Line 374 I think the ACSM guidelines need to appear earlier in your paper either in the introduction or methods section where you describe the LSI index <24> meaning insufficient activity so it would be important for the reader if you acknowledge which ACSM guidelines you are basing this on sooner.

Reply: we have now mentioned the ACSM guidelines in the methods explaining which guidelines we are basing our classification: this has been added: ……

Based on their LSI, patients are classified as active (if LSI ≥24) or insufficiently active (if LSI <24) according to the 2010 release of American College of Sports Medicine (ACSM) Exercise Guidelines for cancer patients [37]. The ACSM guidelines suggest cancer patients engage in at least 150 minutes/week of moderate or 75 minutes/week of vigorous exercise [37].

Reviewer 2 Report

Dear authors,

Please consider incorporating to the manuscript the following suggestions:

1 – Title: The authors might consider changing the title to better reflect the manuscript. Some suggestions: “Exercise intensity and preferences in …” or “exercise levels and preferences in…:

2 – Entire manuscript: Avoid using personal pronouns

3 - Participants and study sample section. For your inclusion/exclusion criteria, did you include participants with active cancer during the time of the data collection? You may want to elaborate on your inclusion/exclusion criteria a little more. Please, explain why you included only participants with past history of cancer and not with active cancer. From your results, it looks that you included participants with past and current history of cancer. Any other exclusion criteria? What about other chronic conditions? Would any other chronic condition influence the outcomes? If so but you didn’t exclude them please list it as a study weakness.

4 - QEX survey (line 87). It is not clear if this questionnaire has been validated.

5 – It would be better if you have the subsection Demographic (line 99) in subsection 2.2 (participants and study sample). Following you can have the outcome measures subsections.

6 – Line 111 to line 119. You referred to reference 27 multiple times, but some of the information from this paragraph does not match reference 27. For instance, you refereed to the ACSM guidelines using reference 27, please refer to the original source. Revise the paragraph for reference accuracy.

7 – Line 346: “We found that patients who underwent chemotherapy were less inclined to participate compared to those who did not.” It would be important to state some possible reasons underlying this finding.

8 – Line 296: “we found that socio-demographic and disease-related characteristics are important modifiers of the willingness to participate”. Please, discuss possible reasons for this finding.  

9 – In general the discussion section needs to be better elaborated. Consider providing possible reasons for all of your findings and compare them to previous literature. 

10 – Line 367: What do you mean by “another possible systematic”?

11 – Although you listed some clinical relevance of the current study in the conclusion you will need to add one paragraph at the end of your discussion listing the clinical relevance of your main findings.

12 – Would past exercise experience interfere with their willingness to participate in an exercise program? For example, someone who had the habit to regularly exercise 10 years before the cancer diagnosis might be more prone to start an exercise program than someone who has never been exposed to exercise.  

13 – Make sure to spell out all the abbreviations when used for the first time

Author Response

Point-by-point reply reviewer 2

Dear authors,

Please consider incorporating to the manuscript the following suggestions:

1 – Title: The authors might consider changing the title to better reflect the manuscript. Some suggestions: “Exercise intensity and preferences in …” or “exercise levels and preferences in…:

Reply: The new title is “Exercise levels and preferences in cancer patients: a cross-sectional study” as suggested. We like the new title! Thank you.

2 – Entire manuscript: Avoid using personal pronouns

Reply: We have modified the text.

3 –a) Participants and study sample section. For your inclusion/exclusion criteria, did you include participants with active cancer during the time of the data collection? You may want to elaborate on your inclusion/exclusion criteria a little more. Please, explain why you included only participants with past history of cancer and not with active cancer. From your results, it looks that you included participants with past and current history of cancer. Any other exclusion criteria?

Reply: The text in the current version can lead to misunderstands and we thank this reviewer for pointing it out to us: All people in the surgery with a past or present history of cancer have been included. Even those that were currently undergoing treatment or that are about to begin (see table 1). More details on the exclusion criteria are now provided and the text between lines 80 and 81 has been remodelled as follows:

Cancer patients' eligibility criteria were: age ≥18 years, a cancer diagnosis and adequate Italian language proficiency to answer the survey questionnaire (QEX). Invited participants included all kinds of cancer survivors (including those whose diagnosis had just been made or was being defined).

3-b) What about other chronic conditions? Would any other chronic condition influence the outcomes? If so but you didn’t exclude them please list it as a study weakness.

Reply: By definition, the patients participating in STIP-ON were sampled to be representative of those attending the Verona oncology clinic (and not the total of patients). Our aim was to investigate the cancer patients who will be available for a future intervention study. The STIP-ON sample therefore naturally excludes patients who, due to the cancer or other chronic or acute conditions, suffer for example from reduced mobility or are not able to complete the questionnaire. Although more severe patients with severe comorbidities are likely to have been excluded from the study, we agree with this reviewer that our patients’ responses may also have been influenced by other comorbidities.

The following sentence has been added to the study limitation:

The patients in STIP-ON were sampled to be representative of those attending the Verona oncology clinic (and not the full total of patients). Therefore, although more severe patients with severe comorbidities are likely to have been excluded, patients’ responses may also have been influenced by other comorbidities that were not investigated by the QEX.

4 - QEX survey (line 87). It is not clear if this questionnaire has been validated.

Reply:  No, there has been no validation against a gold standard or calculation of the repeatability of the information requested.  However, the various items were taken from similar previously published studies. We have now added in the Methods a detailed description of the sources of the various QEX sections.

5 – It would be better if you have the subsection Demographic (line 99) in subsection 2.2 (participants and study sample). Following you can have the outcome measures subsections.

Reply: We have modified the method subsections.

6 – Line 111 to line 119. You referred to reference 27 multiple times, but some of the information from this paragraph does not match reference 27.  For instance, you refereed to the ACSM guidelines using reference 27, please refer to the original source. Revise the paragraph for reference accuracy.

Reply: Thank you. We have now checked and corrected the text and references. 

The new text is as follows:

Questions 10-11:  Level of physical exercise

The QEX inquiry about current exercise level was based on questions from the Godin Leisure-Time Exercise Questionnaire (GLTEQ) [35] which is widely used for cancer patients [36]. A detailed description of the computation of LSI from GLTEO is found elsewhere [35, 36]. In brief: i) The GLTEQ enquires about the previous week’s leisure time frequency (times/week) of vigorous, moderate and mild intensity exercise; ii) Each exercise intensity is associated with the metabolic equivalent of the task (MET): MET = 9 for vigorous, MET = 5 for moderate, MET = 3 for mild intensity exercise [35];  iii) The LSI is then calculated as the sum of (vigorous * 9) + (moderate * 5) per-week exercise frequency according to Godin and Shepard [35]. Based on their LSI, patients are classified as active (if LSI ≥24) or insufficiently active (if LSI <24) according to the 2010 release of American College of Sports Medicine  (ACSM) Exercise Guidelines for cancer patients  [37]. The ACSM guidelines suggest cancer patients engage in at least 150 minutes/week of moderate or 75 minutes/week of vigorous exercise [37]. The QEX includes an additional self-rated question about the frequency (times/week) of sweat-inducing activity. There are three categories of frequency (often/sometimes/never-rarely) These questions and categorization are also taken from GLTEQ  [35].

7 – Line 346: “We found that patients who underwent chemotherapy were less inclined to participate compared to those who did not.” It would be important to state some possible reasons underlying this finding.

Reply: the following have been added in the discussion.

Chemotherapy was inversely associated with the willingness to participate. There is one study that found no relation between cancer treatment and adherence in high-intensity and low-to-moderate intensity exercises [45]; other studies found chemotherapy [44] and its side effects [22] were associated with low adherence to physical exercise programs. One explanation for these contradictory results may be that chemotherapy is a generic term that includes different drugs and various possible side effects.

8 – Line 296: “we found that socio-demographic and disease-related characteristics are important modifiers of the willingness to participate”. Please, discuss possible reasons for this finding.  

Reply: The Discussion has been completely reorganized and the main results are discussed in an orderly manner in relation to the literature. Here's how this part was rewritten:

Several socio-demographic characteristics were associated with the willingness to participate in an exercise program. Willingness decreased with age, also in fully adjusted models, and this was to be expected given the growing difficulties and comorbidities due to ageing.  Age has been associated with low adherence to exercise in cancer patients in various studies [30, 43]. What is interesting is that even among the older patients more than two-thirds said they might be interested in taking part in an exercise program.

Women were more willing to participate than men. That was found in all models, even after adjustment for medical and socio-demographic variables. That women cancer patients adhere better than men in exercise programs is suggested by an intervention study in rectal cancer patients [43] although a systematic review evaluating the predictors of adherence to exercise interventions during cancer treatment suggested that adherence was best among men [44]. Better educated patients were more willing to participate. This was reported in other studies too [30, 45] and a likely explanation is well-educated people’s greater awareness and knowledge of the benefits of exercise.

9 - In general the discussion section needs to be better elaborated. Consider providing possible reasons for all of your findings and compare them to previous literature.

Reply: The discussion has been completely reorganized to make it more incisive.

10 – Line 367: What do you mean by “another possible systematic”?

Reply: that was a typo, we apologise. The new phrase is by  

“Another potential source of error is selection bias: “

11 – Although you listed some clinical relevance of the current study in the conclusion you will need to add one paragraph at the end of your discussion listing the clinical relevance of your main findings.

Reply: We have added the following paragraph at the end of the discussion:

Information from this survey is clinically relevant and may help in designing personalized interventions so cancer patients will achieve sufficient exercise/PA. Here are a few examples: i) Since about 90% of participants said they wanted or needed a helper during the program, a targeted intervention program should include specific activities (and support) for helpers patients will nominate; ii) Because about 30% of respondents said they prefer to exercise with other patients, exercise classes specifically for them and "learning from peers" social occasions should be organized;   iii) The majority of patients were insufficiently active and preferred mild exercise or slow walking. So as not to leave anyone behind, for those who are not able to engage in moderate exercise, a mild flexible entry program should be offered according to the patient’s condition and preferences and then progress slowly towards higher-intensity exercise.  In conclusion, an exploratory survey like STIP-ON could serve as a necessary first step in developing lifestyle improvement interventions for patients. This is particularly important in a country like Italy where there is little knowledge in this field, and factors such as the family environment and social support are not well understood.  Only a small proportion of patients were sufficiently active, although the majority were willing to start an exercise program. Exercise preferences in cancer patients tended to vary substantially. These findings underline the urgency of promoting personalized exercise intervention programs among Italian cancer patients.

12 – Would past exercise experience interfere with their willingness to participate in an exercise program? For example, someone who had the habit to regularly exercise 10 years before the cancer diagnosis might be more prone to start an exercise program than someone who has never been exposed to exercise.  

Reply:  Thanks for the comment. This information was not collected by the QEX; the main reason was to limit the number of QEX items so as not to overburden patients (20 minutes maximum for compilation). We have now noted the absence of this information among the limits of the study and have added the following in the Discussion:

The QEX does not collect information about participants’ pre-diagnosis exercise and physical activity and that limit its ability to explore associations with other possible determinants of current exercise behavior.

13 – Make sure to spell out all the abbreviations when used for the first time

Reply: We have checked the text to make sure that all abbreviations are spelled out when used for the first time.

Reviewer 3 Report

The study deals with an important topic. In order to develop tailored intervention  for cancer patients, it is important to understand what increases participation and adherence in sports and what reduces it.
This involves both psychological predictors, e.g. self-efficacy, control expectations, competence expectations, as well as the relationship between intention and behaviour, but also organisational circumstances, barrieres
How is the intention to be active in sports formed, can specific preferences be assigned to specific groups and why is this so?
I miss the theoretical framing of the study in this paper, e.g. theory of planned behaviour.

I think that the study shows interesting results, but they can hardly be generalized without a theoretical framework. Why is it surprising that there is a heterogeneity in the preferences? The authors argue that the added value also lies in the fact that this is the first study with Italian patients. That is certainly true, but what is the peculiarity, here there is a lack of explanation.

Author Response

Point-by-point reply reviewer 3

The study deals with an important topic. In order to develop tailored intervention  for cancer patients, it is important to understand what increases participation and adherence in sports and what reduces it.
This involves both psychological predictors, e.g. self-efficacy, control expectations, competence expectations, as well as the relationship between intention and behaviour, but also organisational circumstances, barriers
How is the intention to be active in sports formed, can specific preferences be assigned to specific groups and why is this so? I miss the theoretical framing of the study in this paper, e.g. theory of planned behaviour. I think that the study shows interesting results, but they can hardly be generalized without a theoretical framework. Why is it surprising that there is a heterogeneity in the preferences? The authors argue that the added value also lies in the fact that this is the first study with Italian patients. That is certainly true, but what is the peculiarity, here there is a lack of explanation.

Reply: We appreciate this point that enables us to put this survey study in the wider context of our research. This STIP_ON survey had the following aims:

  1. To understand the size of the problem that is: to calculate the prevalence of potentially risky behavior (i.e. insufficient exercise or level) in cancer patients.
  2. To analyze the patients’ characteristics associated with insufficient exercise.
  • To analyze the patients’ characteristics associated with their motivation/willingness to take part in a future intervention program on exercise.
  1. To describe patients’ preferences about exercise.

Information from this survey is helping us design a pilot study to verify the feasibility of a personalized intervention so cancer patients will achieve sufficient exercise/PA. The survey findings help us build the intervention's logic model. Here are a few examples:

  1. Because about 90% of our participants said they wanted or needed a helper during the program, the intervention will include specific activities (and support) for the helpers patients will nominate. (In this case, the component of the logic model will be to "encourage social support".)
  2. Because about 30% of respondents said they prefer to exercise with other cancer patients we are organizing exercise classes specifically for them and there will be moments for sociability. (In this case, the component of the logic model will be "learning from peers".)
  3. Another important item of information concerns the fact that the majority of patients are sedentary and prefer mild exercise or slow walking. We know exercise guidelines for cancer patients suggest they should engage in at least moderate exercise (REF #58 Campbell KL Med Sci Sports Exerc 2019). So as not to leave anyone behind, for those who are not able to engage in moderate exercise, a mild flexible entry program will be offered according to the patient’s condition and preferences and then progress slowly towards higher-intensity exercise. (In this case, the component of the logic model is the ability to draw up a feasible action plan together with the patient)

We thank rev3 for giving us the opportunity to describe our research. Nevertheless, because all this concerns a future phase of our research which is still being developed we felt it was not appropriate to give full details in this article. However, we have added a brief description of the relevance of this study to a subsequent intervention in both the introduction and the discussion:

Introduction

In order to  overcome this information gap the STIP-ON (Sustainable training in pazienti oncologici) survey was designed with the following aims: i) To understand the size of the problem, i.e.: to calculate the prevalence of insufficient exercise among cancer patients;  ii) to analyze the patients’ characteristics associated with insufficient exercise; iii) to analyze the patients’ characteristics associated with their motivation/willingness to take part in a future intervention program on exercise; iiii) to describe patients’ preferences about exercise.

Discussion

Information from this survey is clinically relevant and may help in designing personalized interventions so cancer patients will achieve sufficient exercise/PA. Here are a few examples: i) Since about 90% of participants said they wanted or needed a helper during the program, a targeted intervention program should include specific activities (and support) for helpers patients will nominate; ii) Because about 30% of respondents said they prefer to exercise with other patients, exercise classes specifically for them and "learning from peers" social occasions should be organized; iii) The majority of patients were insufficiently active and preferred mild exercise or slow walking. So as not to leave anyone behind, for those who are not able to engage in moderate exercise, a mild flexible entry program should be offered according to the patient’s condition and preferences and then progress slowly towards higher-intensity exercise.

Round 2

Reviewer 2 Report

The authors did a great job incorporating the suggestions into the manuscript. The manuscript have significantly improved. 

Author Response

We agree with this reviewer that the manuscript has significantly improved thanks to the proactive comments and suggestion received. Thank you

Reviewer 3 Report

The authors have addressed my concerns completely and answered them comprehensively.
Basically, I find it very worthwhile that a pre-survey is carried out. Only then is it possible to carry out a targeted intervention. As this initially results in a pilot project, a well thought-out scientific procedure is available here.
From my point of view, since this is actually a preliminary survey for a pilot study, I would rather classify it as a brief report.

Despite that, I enjoyed reading the paper and suggest a publication.

Author Response

We thank this reviewer for the kind and supportive words. We are convinced that the manuscript has significantly improved thanks to reviewers' proactive comments and suggestions